# Bacon, Brownie, or Broccoli? Beliefs about Stress-Relieving Foods and Their Relationship to Orthorexia Nervosa

**DOI:** 10.3390/nu14183673

**Published:** 2022-09-06

**Authors:** Annebirth Steinmann, Alea Ruf, Kira F. Ahrens, Andreas Reif, Silke Matura

**Affiliations:** Department for Psychiatry, Psychosomatic Medicine and Psychotherapy, University Hospital Frankfurt-Goethe University, 60528 Frankfurt, Germany

**Keywords:** orthorexia nervosa, nutritional beliefs, stress eating, comfort food, eating disorder

## Abstract

Background: Nutritional beliefs play an important role when it comes to food choice. However, little attention has been paid to which foods individuals believe to be comforting when experiencing stress. With increasing health awareness in the general public, this study aims to examine whether the nutritional belief exists that only healthy foods relieve stress. If so, we are interested in its relationship to Orthorexia Nervosa (ON) tendencies. Methods: 175 participants (mean age 28.5 ± 7.8 years, 124 females) completed questionnaires to assess beliefs about stress-relieving foods and ON tendencies. Principal component analysis was used to reduce foods to food groups. Subsequently, a latent profile analysis was performed to identify groups with distinct nutritional beliefs. Results: Among eight distinct groups, one group (8% of the sample) reported the belief that exclusively healthy foods relieve stress. Multinominal logistic regressions showed that higher ON tendencies were associated with that group. Conclusions: Our findings suggest that individuals with stronger ON tendencies believe that, in particular, healthy foods relieve stress. This indicates that nutritional beliefs in ON concern not only the somatic consequences of certain foods, but also psychological consequences, which might also drive orthorexic behaviour. This offers a new target for the diagnosis and treatment of ON.

## 1. Introduction

### 1.1. Nutritional Beliefs

Nutritional beliefs play an important role in everyday life as they influence which foods we buy and eat [1]. Nutritional beliefs include beliefs about ideal dietary patterns, such as when it is best to eat or whether three large meals or several small meals a day are better. Moreover, nutritional beliefs often concern the health impact of certain foods, i.e., which foods are healthy and which are unhealthy [2]. These beliefs are reinforced by a broad spectrum of research that focuses on the somatic impact of different dietary patterns, the mass media’s interaction with it and food industries’ interests [2].

### 1.2. Nutritional Beliefs about Comfort Foods

Nutritional beliefs do not only concern what and how much we eat, but also how foods influence us psychologically. Foods that are consumed with the intention to increase well-being or to dilute negative emotions are often referred to as comfort foods [3]. Although there is a broad range of research on comfort foods and beliefs about foods that can give comfort, there is no clear definition of comfort foods [4]. The belief about which foods are comforting varies across cultures and individuals. In Western society, the image of a hot soup on a couch on a rainy day appears very cosy and generates an image of comfort [4]. In the UK, black tea was rated as one of the top comfort foods [3]. On an individual level, age as well as gender influence what is considered comforting. Moreover, it could be shown that favourite dishes from childhood had comforting effects [3,4]. Thus, what someone believes to be comforting is influenced by associations of a time when parents or caregivers in general provided food [5].

### 1.3. Comfort Foods and Stress Relief

Comfort foods influence how we feel and are consumed to increase well-being. One major risk factor that impacts well-being in everyday life is the experience of stress. It has been shown that eating is used to comfort in stressed states [6,7]. Hence, some individuals use eating in order to relieve stress and increase well-being. Although the body of research on the relationship between stress and eating behaviour is large, it has not been examined which foods are believed to relieve stress and give comfort in stressed states. Instead, studies have mainly focused on what or how much is eaten. It could be shown that greater stress was accompanied by a greater drive to eat, even leading to binge-eating episodes [8]. Furthermore, stress is associated with an increase in high-sugar high-fat (HSHF) intake and a decrease in the consumption of low-energy high-nutrient foods (i.e., healthy foods), particularly fruits and vegetables [9,10]. Individuals with chronic stress are especially prone to consume more calories [11]. This is intriguing, as several studies reported that eating low-energy high-nutrient foods, such as fruits, vegetables, nuts or legumes, can increase overall psychological well-being [12,13,14]. Ecological momentary assessment (EMA) studies confirm these results, showing that fruit and vegetable consumption is associated with feeling happy within the subsequent two hours, but also feeling happier and less sad in general [15,16].

### 1.4. The Role of Healthy Foods in Stress Relief

It seems counterintuitive that stress is mostly associated with unhealthy eating, while it appears that healthy foods increase well-being. An increase in health awareness has been consistently observed [17]. Beyond that, more attention is paid to the psychological effects of food. Finch et al. [18] showed experimentally that healthy comfort eating had the same stress-reducing effect as unhealthy comfort eating [18]. This and the previously mentioned findings build the foundation for our approach: The first aim of our study is to examine whether a group of individuals can be identified that is driven by the belief that healthy foods relieve stress and thus differs from the common assumption that HSHF intake provides comfort when experiencing stress.

### 1.5. Orthorexia Nervosa and Beliefs about Stress-Relieving Foods

In the context of the belief that healthy foods relieve stress, Orthorexia Nervosa (ON) seems particularly relevant, as it is the obsession with eating only foods recognised as healthy because of excessive health concerns [19]. The concept of ON is still rather new and subject to dispute. It is not yet defined as a mental disorder in the common disease classification systems (ICD-11 [20] and DSM-V [21]). However, it is discussed whether ON should be classified as a separate eating disorder [22]. So far, it has been reported that nutritional beliefs in individuals with ON concern mainly somatic consequences, such as that the intake of healthy foods prevents the development of severe somatic conditions [22]. So far, little is known about whether beliefs concerning psychological consequences of food choice also drive orthorexic behaviour. Consequently, nutritional beliefs about stress-relieving foods have not been investigated with regard to ON tendencies yet. We expect the nutritional belief that healthy foods relieve stress is more common in individuals with higher ON tendencies.

To the best of our knowledge, our study is the first to investigate nutritional beliefs about stress-relieving foods and their association with higher ON tendencies. We assess whether a group characterised by the nutritional belief that only healthy foods can help to relieve stress can be identified. If so, this group will be—in accordance with the results of comfort food research—categorised demographically (average age, gender, education, diet and wish for body weight change). We hypothesize that this group is associated with higher levels of ON tendencies.

## 2. Materials and Methods

### 2.1. Participants

Data for this study were collected within the APPetite study [23], which is part of the European Union Horizon 2020 project Eat2beNICE. In total, 124 females and 51 males between the ages of 18 and 53 years (*M* = 28.5 years, *SD* = 7.8) were included in the study. The average BMI was 24.38 (*SD* = 3.95). Participants included healthy adults that were able to give informed consent. Lifetime diagnosis of bipolar I disorder, schizophrenia, organic mental disorder or substance dependence (other than nicotine dependence), current severe episode of other Axis 1 mental disorders, as well as severe somatic conditions were exclusion criteria.

### 2.2. Procedure

Participants who agreed to take part in the APPetite study were invited to complete digitalized questionnaires as part of a longer appointment on campus. Weight and height were measured to calculate Body Mass Index (BMI = weight (kg)/height (m)^2^).

### 2.3. Measures

#### 2.3.1. Nutritional Beliefs about Stress-Relieving Foods

Nutritional beliefs about stress-relieving foods were assessed through a 29-item questionnaire that was developed for the present study. Participants’ beliefs were captured by asking how much each of the 29 foods (see Table 1) helps them to feel better when feeling stressed. The items were rated on a 5-point scale ranging from 1 (*not at all*) to 5 (*very much*). Due to the lack of a commonly used list of food groups suitable for the German population, the list of food items was developed for the present study. To do so, we (1) selected the most relevant food items from the food categories provided by the Competence Center for Nutrition of the Bavarian State Ministry of Food, Agriculture and Forestry [24]; (2) in some cases, split the groups to allow a more detailed assessment (e.g., rice and potatoes); and (3) added missing foods (e.g., plant-based milk/yoghurt) to account for recent developments in the food market, especially the trend towards plant-based foods.

#### 2.3.2. Orthorexic Tendencies

The 10-item Düsseldorfer Orthorexie Scale [25] was used to capture ON tendencies. The DOS has been shown to have good psychometric properties, with Cronbach’s α = 0.84 and a high test–retest reliability of r = 0.79 [25]. It was recently reported that the DOS is a reliable self-report instrument [26]. Items are answered on a 4-point Likert scale ranging from 1 (*does not represent myself*) to 4 (*this represents myself*). Total scores of 30 and above indicate pathological ON tendencies.

### 2.4. Statistical Analysis

Analyses were conducted using R (version 4.1, Vienna, Austria) and RStudio (version 1.4.1106, Boston, MA, USA). Significance was tested at 5% level.

#### 2.4.1. Data Exclusion

All food groups that were rated 1 by at least 60% of participants were excluded, as this indicates that these foods are less relevant in the context of stress relief. The items “Sauces”, “Vegetarian spread”, “Flaxseeds/chia seeds”, “Meat substitute”, “Plant-based milk/Yoghurt” and “Vegetable oil” were excluded due to this criterion (see Appendix A).

#### 2.4.2. Principal Component Analysis

Principal component analysis (PCA) was performed with the R-package *psych* [27] on the remaining 23 food items. In this study, PCA was used as a data reduction method. It allowed the grouping of items into factors and thus revealed latent constructs on the item level. To check the pattern of relationships as an initial data preparation, a correlation matrix including all correlations between all food items was examined (see Appendix A). Model adequacy was assessed via Kaiser–Meyer–Olkin (KMO) [28] measure, Bartlett’s test [29] and minimum covariance determinant [30]. For the final PCA, 19 remaining food items were used. As recommended as the best method for factor extraction, parallel analysis was applied [31]. Orthogonal rotation was chosen to maximize the loading on only one factor for each variable. For further person-centred analysis, factor scores were calculated using the Anderson–Rubin method—a modification of the Bartlett’s method—adaptable for orthogonal factor solutions only (see Appendix A).

#### 2.4.3. Latent Profile Analysis

Latent profile analysis (LPA) is a person-centred clustering approach that uses continuous variables as indicators. LPA was executed with the R-package *tidyLPA* [32]. The most complex model—usually the recommended model [33]—with regards to the constraint of class-specific (co)variance matrices of the indicator variables was configured [32]. Various fit indices are available to determine the number of latent profiles [34]. As recommended, estimation outputs were inspected initially for error messages, outliers and theoretical plausibility [35]. Subsequently, we compared the BLRT, SABIC, BIC and AIC. According to the literature, the BLRT (bootstrap likelihood ratio test) is seen as one of the most accurate fit indicators [36]. In particular, BLRT efficiently selected the correct number of latent profiles when outcome variables were nonnormal [37]. Thus, in model selection, we focused firstly on a significant BLRT. Another accurate fit indicator is the SABIC (sample-size adjusted BIC), which is corrected when compared to the BIC (Bayesian information criterion) [35]. The informative value of the AIC (Akaike information criterion) is, compared to the BIC and SABIC, considered to be weaker [38]. However, the performance of AIC is superior in selecting the correct number of profiles with sample sizes <500 [35]. The entropy quantifies separation, with values >0.8 being taken to indicate sufficient separation. In case of a disagreement of fit indices on the optimal number of profiles, the profile sizes of the different profile solutions were considered. The literature recommends that no profile should be <5% of the sample [38]. To further characterize the profiles, means and standard deviations were calculated for age and BMI. Moreover, sex distribution of the profiles was compared.

#### 2.4.4. Multinominal Logistic Regression Analysis

Multinominal logistic regression analysis (MLRA) was applied to investigate the association between higher ON tendencies and nutritional beliefs about stress-relieving foods. The number of covariates used in MLRA depended on the group sizes of profiles formed through LPA. Beta values, their standard errors and odds ratios (OR) in combination with the corresponding 95% confidence intervals (CI) are given as estimates of effect sizes.

## 3. Results

### 3.1. Sociodemographic Characteristics

The sociodemographic characteristics of the sample are summarized in Table 2. In total, 124 females and 51 males between the ages of 18 and 53 (*M* = 28.5 years, *SD* = 7.8) were included in the study. Female participants were on average 28.1 (*SD* = 7.7) and male participants were on average 29.7 (*SD* = 8.1) years old. The average BMI of the total sample was 24.38 (*SD* = 3.95). The average BMI of female participants was 23.60 (*SD* = 3.68). Male participants had an average BMI of 26.31 (*SD* = 3.96). Within the sample, 40% had graduated from high school (Abitur: German equivalent of “A Levels”), 22% were bachelor’s graduates and 23% were master’s graduates. Moreover, 5% reported having accomplished their Ph.D., and another 5% had completed vocational training. Only 1% reported having an intermediate school-leaving certificate (mittlere Reife: German equivalent of “high school diploma”).

#### Diet

A pescetarian diet was followed by 5% of the sample, whereas 11% were vegetarians. Only 3% reported a vegan diet. The majority of the sample (81%) followed an omnivore diet. Regarding restricted diet to lose weight, 48% of the sample reported that they were trying to reduce their weight, whereas 3% reported that they aimed to gain weight. The remaining 49% reported that they did not regulate their weight at all.

The mean DOS score of the sample was 17.77 (*SD* = 5.18).

### 3.2. Principal Component Analysis

KMO measurement (KMO = 0.88) was “great” according to Kaiser [39]. This as well as the significant Bartlett’s test verified the sampling adequacy for the PCA. The minimum covariance determinant (MCD) was just above the cut-off value of 0.00001 (MCD = 0.0000341). There was no multicollinearity (bivariate correlation coefficients > 0.9) present (Appendix A). A first PCA was calculated (see Appendix A). All items that scored on more than one factor with similar factor loadings were excluded. This mainly affected food items rich in carbohydrates (“Bread”, “Rice”, “Potatoes”, “Pasta”). PCA was again calculated with 19 remaining variables (see Appendix A). Model adequacy parameters verified the sampling adequacy (KMO = 0.83 and Bartlett’s test was significant (*p* < 0.05)), and the determinant was sufficient (MCD = 0.00075). Parallel analysis estimated three components to be the precise number of components. The model explains 53% of the variance. A cut-off of 0.4 was considered for factor loadings. Items with factor loading higher than 0.4 were included. Inspection of the factors led to labelling of “animal-based products” (Factor one), “junk food” (Factor two) and “healthy vegetarian products” (Factor three).

### 3.3. Latent Profile Analysis

Profile enumeration was calculated from the most complex model, in which the variances and the covariances were freely estimated across profiles [32]. Fit indices for the three- to eight-profile solutions, listed in Table 3, disagreed on the optimal number of profiles. The BLRT revealed significant results in the three-profile- and eightprofile-solution, whereas the BIC was best in the three-profile solution. The SABIC and AIC were lowest in the six-profile solution, but second lowest in the eight-profile solution. Entropy showed sufficient profile separation in the three- and six- to eight-profile solutions. Thus, in order to determine the optimal profile solution, entropy was not useful, as all potential solutions had values >0.8.

As a significant BLRT outperforms other fit indices [34], we initially focused on this value. However, because AIC and SABIC were smallest in the six-profile solution, we decided to estimate profile sizes of the six- and eight-profile solutions to finally choose the optimal number (Table 4). Based on this procedure, the six-profile solution was rejected. Here, the smallest profile includes only five members. This violates the recommendation that profile size should be at least 5% of the sample [38]. Hence, the model with eight profiles was chosen to be the best model due to its significant BLRT, second best AIC and SABIC, and sufficiently large profile sizes, but also regarding its content.

Eight profiles emerged from the LPA, presented in Figure 1. Based on the mean factor scores, labels were chosen to describe the nutritional beliefs about stress-relieving foods of each profile. Profile 1 (*n* = 37) believed that junk as well as healthy foods relieve stress. Thus, we named it according to the participants described in this profile: “Junk and veggie believer”. Furthermore: Profile 2: “Low junk believer” (*n* = 39); Profile 3: “Non-believer” (*n* = 13); Profile 4: “Strong animal-product believer” (*n* = 20); Profile 5: “Junk and animal-product believer” (*n* = 10); Profile 6: “General believer” (*n* = 19); Profile 7: “Low animal-product believer” (*n* = 23); Profile 8: “Healthy believer” (*n* = 14). For instance, Profile 2 was named “Low junk believer” according to a negative mean value of the factors animal and healthy foods and a low positive mean value of the factor junk food. The profiles differed regarding age (see Table 5). Individuals that belonged to Profile 8 (“Healthy believer”) and Profile 6 (“General believer”) were on average 32 years old and, therefore, the oldest groups. Profile 5 (“Junk and animal-product believer”) (26.4 ± 6.43) represented the youngest group. The profiles showed no large differences in BMI, as all profiles showed normal average weight except for Profile 6 (“General believer”) (26.7 ± 5.5). Regarding sex distribution, LPA revealed that female sex predominated in most profiles (see Table 5). Exceptions were Profile 4 (“Strong animal-product believer”) and 6 (“General believer”), in which sex was almost equal, and Profile 7 (“Low animal-product believer”), in which male sex predominated.

### 3.4. Multivariate Logistic Regression

To test whether higher ON tendencies could predict membership to a certain profile, MLRA was performed. Profile 3—the profile of “Non-believer”—was chosen as a reference group. This profile was suitable as baseline because members did not believe any foods relieve stress. Table 6 shows that participants with higher ON tendencies were significantly more likely to be a member of the profile (Profile 8—Healthy believer) that only believed healthy foods relieved stress (*OR* = 1.28, *CI* = [1.07; 1.53], *p* < 0.01) compared to the reference group of “Non-believer”. Participants with higher ON tendencies were also significantly more likely to be a member of the “Junk and veggie believer” profile (*OR* = 1.24, *CI* = [1.05; 1.46], *p* < 0.05). Thus, higher levels of ON tendencies predict profile membership to the profiles with the nutritional belief that either only healthy or healthy and junk foods relieve stress.

## 4. Discussion

The aim of the present study was to examine whether (a) the nutritional belief exists that only healthy foods relieve stress, and (b) this belief is associated with higher ON tendencies.

First, we identified a group of individuals with the nutritional belief that healthy (i.e., whole-food-plant-based) foods, such as fruits, vegetables, salad, nuts and soups, help to relieve stress. Confirming our first hypothesis, higher ON tendencies were associated with this belief. Unexpectedly, the group with the nutritional belief that both healthy and junk foods, such as cakes, sweets, sweet spreads and salty nibbles, are stress-relieving was also associated with higher ON tendencies.

The nutritional belief about HSHF stress-relieving foods has been previously shown to be the most prevalent [40,41]. However, our results show that other nutritional beliefs about stress-relieving foods exist, namely, the nutritional belief about healthy stress-relieving foods. In comfort food research, 40% of what is reported as comforting could be categorized as homemade, natural or even as healthy [4]. Our results show that nutritional beliefs about stress-relieving foods are similarly heterogenic to nutritional beliefs about comfort food. Interestingly, we were able to identify a group of individuals who reported the nutritional belief that only healthy foods are stress-relieving.

Interestingly, the group of “Healthy believers” (i.e., the group with the nutritional belief that healthy foods are stress-relieving) showed demographical characteristics that were also reported in research on comfort foods. Thus, age and gender seem to influence the preferences for stress-relieving foods. For instance, it has been reported that the need for sweet comfort food decreases with age [42]. Simultaneously, older individuals report choosing healthier comfort food in general [4]. Our results on nutritional beliefs about stress-relieving foods are consistent with these observations. Compared to the other groups, participants of the “Healthy believers” group were older. A national survey in Germany reported that the interest in health information increases with age [17]. It can be suggested that this is related to a behaviour that is characterized by a higher health awareness and different nutritional beliefs regarding stress-relieving foods. Furthermore, women predominated in the group of “Healthy believers”. In comfort food research, it has been shown that gender has an impact on the choice of comfort foods [3]. Women have been reported to live healthier in general [43] and are often more aware of what is considered healthy [17].

Our hypothesis that individuals with higher ON tendencies believe that healthy foods can relieve stress could be confirmed. Previous research on ON has mainly focused on nutritional beliefs concerning the somatic consequences of nourishment [22]. Thus, our study extends the research on nutritional beliefs in individuals with higher ON tendencies. Nutritional beliefs play an important role in the development and maintenance of eating disorders, as they are the foundation for strict rules and the adherence to them [40]. Individuals profit from this pathological behaviour, as it is comforting and, in the case of ON, consuming healthy foods contributes to the feeling “to do everything right” [22]. Stress is often accompanied by negative feelings. Thus, in stressed states, it may be even more important to stick to orthorexic behaviour. Indeed, it could be shown that stress aggravates symptoms in patients with eating disorders. Bulimia Nervosa as well as Binge-Eating patients are triggered into binge-eating episodes when stressed [44], whereas Anorexia Nervosa patients decrease food consumption. ON is characterized by highly restrained eating patterns with inflexible dietary rules and compulsive behaviours [45]. Additionally, healthy foods are idealised in a way that they are attributed with exaggerated benefits. Consequently, it is not surprising that participants with higher ON tendencies believe that sticking to healthy eating behaviour can relieve stress. Further studies are required to prove these assumptions.

Notably, individuals with higher ON tendencies did not only belong to the group of Healthy believers. There was a small group of individuals with higher ON tendencies reporting that they find both healthy and junk food stress-relieving. Demographically, the group of “Junk and veggie believers” was on average younger than the group of ”Healthy believer”. This result confirms findings from earlier studies, as the need for sweets decreases with age [42]. However, given the association between ON and restrained eating [46], this finding also poses the question whether individuals with higher ON tendencies who believe only healthy products relieve stress have more inflexible dietary rules compared to those who believe both healthy and unhealthy foods relieve stress. ON is not listed as a psychiatric disorder in the existing diagnostic systems (ICD-11 [20] and DSM-V [21]). Particularly, the severity of symptoms associated with ON that would classify it as an eating disorder is still being discussed [45,47]. Thus, our results confirm the need to extend the research on ON. Further research is needed to understand who reported the nutritional belief that only healthy foods relieve stress and who reported the nutritional belief about junk and healthy stress-relieving foods.

Our study extends previous research in that a person-centred approach (i.e., LPA) was used to define different nutritional beliefs about stress-relieving foods. To the best of our knowledge, this is the first study to explore nutritional beliefs about stress-relieving foods and their relation to ON.

However, our study has some limitations: (1) The sample of our study is not representative of the general population, as it entails an overrepresentation of female, younger and well-educated participants. University students are quite a homogenous group of people. They are easy to recruit, potentially having a higher economic motivation to participate in studies and also having sufficient time to take part in studies. (2) There are thousands of food products on the market. The countless number of foods makes the assessment of food-related constructs (such as beliefs about stress-relieving foods) difficult. Unfortunately, to the best of our knowledge, there is no commonly used list of food items suitable for the German population available. Therefore, food items had to be selected for the present study. On the one hand, including all available foods in the assessment was not feasible. On the other hand, only including very broad food categories could oversimply the matter. Therefore, we strived to strike a happy medium. However, the selection of the food items might have directly influenced the results of the present study. Therefore, the results should be interpreted with caution. (3) In person-centred approaches—such as LPA—sample size is crucial for statistical power. With a rather small sample size (N = 175), our study is at the lower bound of the recommendations [35]. For LPA, a power analysis is not necessarily expected [35], and previous studies revealed valid LPA results with smaller sample sizes [48]. However, in order to replicate the profile solution in future studies, we suggest performing a Monte Carlo simulation in advance to calculate the required sample size as recommended by experts in the field [49,50]. Furthermore, MLRA’s power is affected by small group sizes. Still, we complied with the recommendations regarding the rule of one predictor per ten observations. A further limitation is the lack of a validation analysis. Unfortunately, our sample size did not allow us to perform multiple-groups LPA for validation as recommended in the literature [35]. Due to capacity reasons, our study lacks the possibility to replicate the results across different samples, contexts and time points. As further research on nutritional beliefs about stress-relieving foods and their relationship with eating disorders is needed, we strongly recommend larger sample sizes for future studies. Furthermore, the plausibility of the profiles should be replicated by adding additional constructs, such as BMI and eating habits.

## 5. Conclusions

Our study provides novel evidence that helps advance our understanding of the underlying belief system of individuals with higher ON tendencies. Our findings suggest that nutritional beliefs in ON concern not only the somatic consequences of certain foods, but also psychological consequences. Hence, beliefs concerning psychological consequences of food choices seem to be a relevant driver of orthorexic behaviour. These findings offer a new target for the diagnosis as well as treatment of ON.

## Figures and Tables

**Figure 1 nutrients-14-03673-f001:**
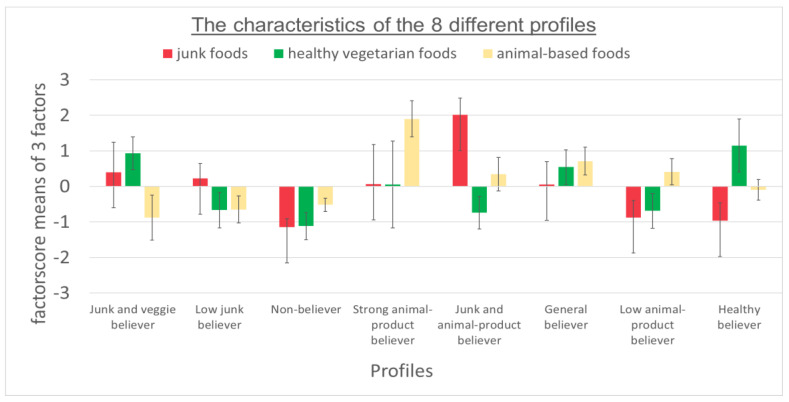
Profiles that were identified in the total sample through latent profile analysis. Error bars represent one standard error.

**Table 1 nutrients-14-03673-t001:** Food items listed in the questionnaire to assess nutritional beliefs about stress-relieving foods.

Food Items Listed in the 29-Item Questionnaire			
Bread	Butter/margarine	Cake	Cereals
Cheese	Couscous/bulgur/pseudocereals	Eggs	Fish
Fruits	Flax- or Chia seeds	Legumes	Meat
Meat substitute	Milk products	Nuts	Pasta
Potatoes	Plant-based milk/yoghurt	Ready meals	Ready sauces
Rice	Salad	Salty nibbles	Soups/stews
Sweet	Sweet spread	Vegetables	Vegetable oil
Vegetarian spread			

**Table 2 nutrients-14-03673-t002:** Sociodemographic characteristics of participants.

Variables		*N* = 175	%
Gender	Male	51	29
	Female	124	71
Education	Mittlere Reife	2	1
	Abitur	71	40
	Completed vocational training	9	5
	Degree of university of Applied Sciences	5	3
	Bachelor’s graduates	39	22
	Master’s graduates	40	23
	Ph.D.	9	5
Diet	Omnivore	142	81
	Pescetarian	9	5
	Vegetarian	19	11
	Vegan	5	3
Body weight change	Yes, I am trying to reduce my body weight	84	48
	No	85	49
	Yes, I am trying to gain body weight	6	3
Age	Male	*M* = 29.7 (*SD* = 8.1)	
	Female	*M* = 28.1 (*SD* = 7.7)	
BMI	Male	*M* = 26.31 (*SD* = 3.96)	
	Female	*M* = 23.60 (*SD* = 3.68)	

**Table 3 nutrients-14-03673-t003:** Fit indices for the profile enumeration.

Profiles	LogLik	AIC	BIC	SABIC	Entropy	BLRT_p
3	−692.284	1442.569	**1534.348**	1442.514	**0.865402**	**0.039604**
4	−680.86	1439.721	1563.147	1439.646	0.779699	0.158416
5	−674.253	1446.506	1601.58	1446.412	0.793919	0.90099
6	−649.797	**1417.594**	1604.316	**1417.482**	**0.868484**	0.079208
7	−659.24	1456.481	1674.851	1456.349	**0.880146**	0.742574
8	**−630.626**	1419.253	1669.271	1419.102	**0.918113**	**0.009901**

AIC = Akaike information criteria; BIC = Bayesian information criteria; SABIC = sample size-adjusted Bayesian information criteria; BLRT = bootstrap likelihood ratio test. Note: Bold font is used to highlight values corresponding to the best-fitting model.

**Table 4 nutrients-14-03673-t004:** Profile sizes of the six- and eight-profile solutions.

Profiles	1	2	3	4	5	6	7	8
Six-profile solution	28	31	28	33	5	50		
Eight-profile solution	33	34	13	10	20	19	23	14

**Table 5 nutrients-14-03673-t005:** Mean scores of indicators used in LPA, group name and size, as well as age, BMI, sex distribution and DOS of the 8 profiles showing 8 different nutritional beliefs about stress-relieving foods.

	Size (*n*)	Sex (m/f)	% of Total (m/f)	Age	BMI	DOS
“Junk and veggie believer”	37	5/32	9.8/25.8	27.84 ± 7.41	23.3 ± 3.58	19.41 ± 5.68
“Low junk believer”	39	9/30	17.6/24.2	28.59 ± 7.12	24.2 ± 3.36	17.51 ± 4.70
“Non-believer”	13	3/10	5.9/8	29.07 ± 8.1	24.4 ± 3.62	15.08 ± 3.04
“Strong animal-product believer”	20	8/12	15.6/9.7	27.25 ± 7.76	25.0 ± 3.7	17.40 ± 4.32
“Junk and animal-product believer”	10	1/9	1.9/7.3	26.4 ± 6.43	23.9 ± 3.52	18.30 ± 6.11
“General believer”	19	8/11	15.7/8.9	32.16 ± 9.56	26.7 ± 5.5	17.68 ± 4.04
“Low animal-product believer”	23	14/9	27.5/7.3	26.43 ± 6.56	23.6 ± 3.12	15.48 ± 6.11
“Healthy believer”	14	3/11	5.9/8.9	32.07 ± 9.7	24.9 ± 5.23	20.64 ± 4.91

**Table 6 nutrients-14-03673-t006:** Odds ratios dependent on DOS total score for being classified into a specific profile with the “NON-believer” profile as reference group.

	DOS Total Score		
	*B* (*SE*)	*OR*	95% *CI*
Non-believer vs.			
Junk and veggie believer	**0.21 (0.09) ***	**1.24**	**1.05; 1.46**
Low junk believer	0.14 (0.85)	1.15	0.97; 1.36
Strong animal-product believer	0.14 (0.09)	1.15	0.96; 1.37
Junk and animal-product believer	0.17 (0.1)	1.19	0.98; 1.45
General believer	0.15 (0.09)	1.16	0.97; 1.39
Low animal-product believer	0.03 (0.09)	1.03	0.86; 1.24
Healthy believer	**0.25 (0.09) ***	**1.28**	**1.07; 1.53**

Asterisks indicate level of statistical significance: * *p* < 0.05; Log-Likelihood: −337.07, McFadden *R*^2^: 0.025971, Likelihood ratio test: χ^2^ = 17.975 (*p* value = 0.012083), bold numbers show significant results.

## Data Availability

The data presented in this study are available on request from the corresponding author. The data are not publicly available due to privacy and ethical concerns.

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
