# Peer review of "Bacon, Brownie, or Broccoli? Beliefs about Stress-Relieving Foods and Their Relationship to Orthorexia Nervosa"

_nutrients, 2022, doi:10.3390/nu14183673_

Round 1

Reviewer 1 Report

In the interesting article entitled "Bacon, brownie, or broccoli? Beliefs about stress-relieving foods and their relationship to Orthorexia Nervosa", the authors revealed that individuals who believe in the stress-relieving effects of healthy foods display higher levels of orthorexic eating behavior. This aspect has not been investigated before, hence, the authors present new and interesting results that increase our knowledge about orthorexic eating behavior. 

In general, the manuscript is well structured and easy to read. Please find here my comments to specific aspects of your manuscript: 

Methods

- line 96: please provide more information about the participants. Which inclusion criteria did you use? What about the level of education and the employment status? Where there any participants following a vegan or vegetarian diet or restricting their diet to lose weight? What about eating disorders? Please also report the mean DOS score for the whole sample. 

- line 111: how were these 29 foods items selected? There are thousands of foods and the choice you present here directly influences the results. Hence, I would like to know which criteria you used to select these foods and why you used a mixture of food groups (e.g. legumes, nuts) and single foods (rice, potatoes). Furthermore, some of the items are directly related (e.g. butter/margarine and vegetable oil, potatoes and vegetables, sweet and cake) while others are not. I assume that the mixture of interdependences and non-related foods (e.g. fruits and meat, rice and cheese) influence the resulting categories and might produce artificial results. 

- since I am not an expert regarding latent profile analysis, I cannot evaluate this part of the manuscript and recommend to have an expert check this. 

Results

- table 4: I suggest to add the mean DOS score for each profile 

Discussion

- line 247: the word "belief" is missing after "nutritional"

To sum up, I believe that this manuscript is very interesting and provides new insights into orthorexic eating behavior. I would like to know more about the choice of food items because this is rather crucial when using these items in a PCA. Hence, I recommend "minor revision" and hope that the authors will be able to provide a good explanation for their choice of foods. Otherwise, these aspects should be mentioned in a limitations section, which is currently missing in the manuscript. 

Author Response

We highly appreciate the time and effort that you have dedicated to reviewing our manuscript titled “Bacon, brownies or broccoli? Beliefs about stress-relieving foods and their relationship to Orthorexia Nervosa”. Thank you very much for the opportunity to submit a revised version of our manuscript. We are grateful for the valuable and insightful comments which we have carefully considered, and which helped us improve the manuscript.

Our point-by-point responses are provided in the attachment. All modifications have been tracked in the manuscript.

We look forward to hearing from you and are happy to respond to any further questions and comments you may have.

Reviewer 2 Report

Overall, the paper was well written, it was clear and concise; and had a thorough analysis of the data. The overall paper has merit to the contribution to the literature with minor revisions. Below are some comments/recommendations to strengthen the paper:

1.       Title of Paper: Consider a different title, the use of bacon, brownie, and broccoli are only used in the title, and seems misleading. These foods were also not included in the food item list in the questionnaire to assess nutritional beliefs about stress relieving foods.  

2.       Line 45-46 needs a reference

3.       Sections 1.3 and 1.4: overall this section is well written but with a focus mainly on “healthy” food, with very little focus on “unhealthy” foods. It is highly recommended the authors expand on the literature for “unhealthy” foods and stress, so the reader can fully understand concepts of stress related to both health and unhealthy foods. Strengthening this section will better align with the discussion.

4.       Methods: in the participate section provide demographics age and average BMI independently for sex (males and females) and not combined.

5.       Methods Measurements: please provide validity and reliability for each of the measures used, and the reliability for your data set.

6.       Methods Statistical Analysis: provide your power analysis to determine whether their sample size is sufficient.  

7.       Results Section: provide how many surveys were completed vs. not used due to incomplete data.

Author Response

(The authors gave the same response as above.)
